


# Evolution of small-scale turbulence at large Richardson numbers

Lev Ostrovsky[1], Irina Soustova[2], Yuliya Troitskaya[2], and Daria Gladskikh[2,3,4]

[1]Dept. of Applied Mathematics, University of Colorado, Boulder, CO, 80309 USA
[2]Institute of Applied Physics, Russian Academy of Sciences, 603950 Nizhny Novgorod, Russia
[3]Moscow Center for Fundamental and Applied Mathematics, 119991 Moscow, Russia
[4]Research Computing Center, Moscow State University, 119991 Moscow, Russia

**Correspondence:** Lev Ostrovsky (lev.ostrovsky@gmail.com)

**Abstract.** The theory of stratified turbulent flow developed earlier by the authors is applied to data from the upper oceanic level to confirm that small-scale turbulence can be amplified and supported at a quasi-stationary level even at large gradient Richardson numbers due to the exchange between kinetic and potential energies. Using the mean profiles of Brunt-Vä is älä frequency and vertical current shear given in Forryan et al. (2013), the profiles of kinetic energy dissipation rate are calculated,

to be in reasonable agreement with the experimental data. This confirms the importance of including potential energy into realistic models of subsurface turbulence.

## 1  Introduction

At present, it is well established that the processes in the upper mixed layer of the ocean and inland waters play a significant role both in the development of global climate models and in the creation of regional weather forecast models (e.g., Hostetler et al.,

1993; Ljungemyr et al., 1996; Tsuang et al., 2001; Mackay, 2006). Small-scale and mesoscale processes effectively interact with each other and provide energy sink for currents and waves of larger scales. Such processes as wind wave breaking, surface and subsurface shear flows created, in particular, by anomalously large surface waves under hurricane winds and intense solitons of internal waves, can cause turbulent mixing and the resulting fine structure formation with areas of sharp gradients of temperature and salinity. Whereas the mechanisms of generation of small-scale turbulence are understood reasonably well,

the problem of its interaction with other types of motions and long-time support is less clear. Earlier works were based on the Miles's instability condition $R_i < 1/4$, where $R_i$ is the gradient Richardson number. However, in many observations, turbulence exists in a quasi-stationary regime at much larger $R_i$, up to 10 and more. In some works, it was explained by the presence of fine structure of current, with thin layers of strong shear (Smyth et al., 2013). A more general description is based on the semi-empirical $k - \varepsilon$ equations (Burchard, 2002; Burchard and Bolding, 2002; Mellor and Yamada, 1982) showing that

the developed turbulence can be amplified and supported under a softer condition $R_i < 1$ (Monin and Yaglom, 1964). Similar equations were used in more specific models of formation of the upper turbulent layer (Ostrovsky and Soustova, 1969), and the action of internal waves on turbulence (Ivanov et al., 1983; Strang and Fernando, 2001; Stretch et al., 2001). However, even that is insufficient to explain many observations in the ocean and atmosphere where the turbulence is observed at significantly



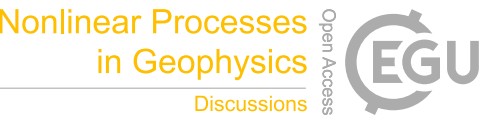

larger $R_\mathrm{i}$, up to 10 and more (Forryan et al., 2013; Avicola et al., 2007; Galperin et al., 2021). New theoretical models have
also appeared for describing non-stationary turbulent processes in the atmosphere and ocean, based on spectral approaches,
confirming, in particular, the absence of a critical Richardson number for describing turbulent-wave processes in a stratified
fluid (Sukoriansky et al., 2003, 2005b, a; Galperin and Sukoriansky, 2020; Galperin et al., 2021, 2007).

In Ostrovsky and Troitskaya (1987), in the framework of a kinetic approach, a closed non-stationary model of turbulence
interacting with a variable current was suggested, that includes a mutual transformation between kinetic and potential energies
of turbulence. The latter is due to the density fluctuations occurring in stratified turbulence (Monin and Ozmidov, 1981).
The theory suggested in Ostrovsky and Troitskaya (1987) is based on a solution of the equation for the variable probability
distribution function of fluid velocity and density. This approach reduces the uncertainty of standard semi-empirical $k - \varepsilon$
schemes and naturally includes the potential energy of turbulence in the model. As a result, small-scale turbulence can be
supported at a non-zero level by the average shear at any finite values of the gradient Richardson number, without a threshold.
Later this approach was further developed in Zilitinkevich et al. (2007a, b); Soustova et al. (2020) in application to atmospheric
turbulence, where the energy and flux budget (EFB) model was added to the theory. The correspondence between this theory
and the $k - \varepsilon$ model, as well as the proper parametrization related to the turbulent Prandtl number, are discussed in recent works
(Gladskikh et al., 2023; Rodi, 1980).

In this paper, the non-stationary kinetic model of turbulence is used to describe the evolution and structure of the upper tur-
bulent layer with the parameters taken from in situ observations. Particular attention is paid to the cases of the large Richardson
number and the role of turbulent potential energy in explaining the observation data. As an example, we use some data from
the paper (Forryan et al., 2013) that provided a relatively detailed set of measured data for three cruises made in 2006-2009
in different areas of the world ocean: North Atlantic (cruise D3406, June-July 2006, and cruise D321, July - August 2007)
and Southern Ocean (cruise JC29, November-December 2008). These experiments were aimed at studying turbulent mixing
in the presence of a stratified shear flow associated with mesoscale motions such as eddies and fronts. With the given profiles
of current shear and buoyancy frequency taken from Forryan et al. (2013), the theory developed in Ostrovsky and Troitskaya
(1987) yields the results that satisfactorily agree with the measurements of the turbulent dissipation rate given in Forryan et al.
(2013). The details of measurements can be found in Forryan et al. (2013) and references therein.

## 2   Basic equations

The general equations obtained in Ostrovsky and Troitskaya (1987), see also Gladskikh et al. (2023), are shown in the Ap-
pendix. Here they will be used for the particular case of known profiles of horizontal current shear $\partial \langle u_x \rangle / \partial z = V_z$ where
$\langle u_x \rangle = V(z)$ is the ensemble-average horizontal velocity, and average density $\langle \rho(z) \rangle = R(z)$. Here $z$ is vertical coordinate. As
a result, we have a system of two equations for the kinetic energy of turbulence $b(t,z)$ and potential energy $P(t,z)$ per unit
volume. The latter is related to the density fluctuations $\langle \rho'^2 \rangle$:

$$P = \frac{\langle \rho'^2 \rangle \mathbf{g}^2}{2N^2 R^2} \tag{1}$$





Here **g** is the gravity acceleration, $R(z)$ is defined above, and $N^2(z)$ is the squared Brunt-Väisala frequency. Note that here $z$-direction is chosen downwards. Under the above conditions, the general equations (18) of Ostrovsky and Troitskaya (1987) or (6) of Gladskikh et al. (2023) are reduced to two equations for $b$ and $P$:

$$\frac{\partial b}{\partial t} = V_z^2 L\sqrt{b} - N^2 L\sqrt{b}\left(1 - \frac{3P}{b}(1-G)\right) - \frac{Gb^{3/2}}{L} + \frac{5}{3}\frac{\partial}{\partial z}\left(L\sqrt{b}\frac{\partial b}{\partial z}\right), \tag{2}$$

$$\frac{\partial P}{\partial z} = N^2 L\sqrt{b}\left(1 - \frac{3P}{b}(1-G)\right) - \frac{Db^{1/2}}{L}P + \frac{\partial}{\partial z}\left(\frac{L\sqrt{b}}{N^2 R^2}\frac{\partial(N^2 R^2 P)}{\partial z}\right).$$

Here $L$ is the outer scale of turbulence and $G$ is the anisotropy parameter tending to 1 for strongly anisotropic turbulence with a small vertical scale compared to the horizontal scale. For details see Ostrovsky and Troitskaya (1987). Here we use the model of locally isotropic turbulence, for which $G \sim 0.5$. The use of a more sophisticated model that accounts for the turbulence anisotropy is redundant for the purposes of this paper, which involves comparison with experimental data (Forryan et al., 2013) that have uncertainties within an order of magnitude. The parameter $L$ can be taken from empirical data (Rodi, 1980) or found from the turbulence spectrum (Forryan et al., 2013; Lozovatsky et al., 2006). $C$ and $D$ are empirical constants. The terms $Cb^{3/2}/L$ and $Db^{1/2}P)/L$ in (2) define the dissipation rates of kinetic and potential energy, respectively.

Before analyzing the full system (2), we note that some significant conclusions can be made from a reduced, local ODE system following from (2) after neglecting the last, diffusive terms in these equations:

$$\frac{\partial b}{\partial t} = V_z^2 L\sqrt{b} - N^2 L\sqrt{b}\left(1 - \frac{3P}{b}(1-G)\right) - \frac{Cb^{3/2}}{L}, \tag{3}$$

$$\frac{\partial P}{\partial z} = N^2 L\sqrt{b}\left(1 - \frac{3P}{b}(1-G)\right) - \frac{Db^{1/2}}{L}P.$$

The coordinate $z$ is now a parameter in these equations. In particular, they define a stationary distribution (a stable equilibrium point on the phase plane of variables $b$ and $P$):

$$b_{\text{st}}(z) = \frac{V_z^2 L^2}{2C}f(R_i), \quad P_{\text{st}(z)} = \frac{V_z^2 L^2}{D} - b_{\text{st}} \tag{4}$$

where $R_i = V_z^2/N^2$ is Richardson number, and

$$f(R_i) = 1 - (4-3G)R_i + \left[1 + R_i^2(4-3G)^2 + R_i(4-6G)\right]^{1/2}. \tag{5}$$

It is noteworthy that at $R_i \to \infty$, $f$ has a non-zero limit $f_\infty = 6(1-G)/(4-3G) > 0$ (it is 1.2 for $G = 0.5$). Hence, the turbulent energy remains finite at large Richardson numbers. General features of this solution and the turbulent Prandtl number following from it are discussed in Ostrovsky and Troitskaya (1987); Gladskikh et al. (2023). Certainly, the applicability of these simple solutions must be verified by solution of full equations (2) or at least by estimates of the diffusion terms neglected in these equations. In the experiments, the dissipation rate of kinetic turbulent energy $\varepsilon$ is commonly measured as a characteristic of turbulence. Within the semi-empirical approach, it is defined as Kolmogorov (1941):

$$\varepsilon = \frac{Cb^{3/2}}{L}. \tag{6}$$




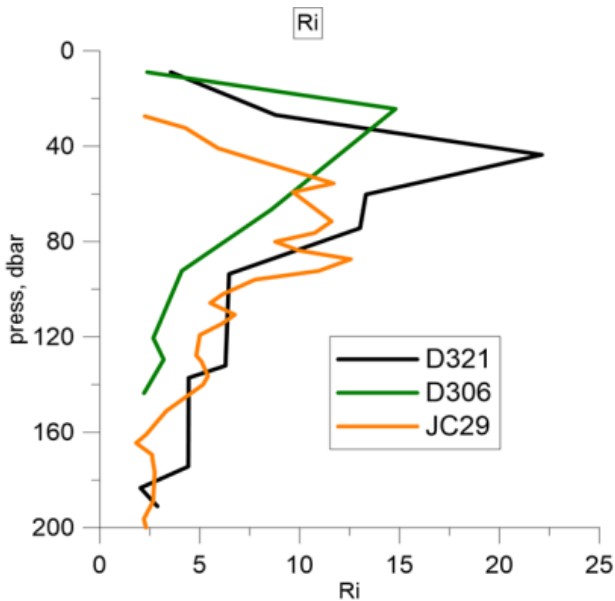

**Figure 1.** Profiles of Richardson number for three cruises calculated from Fig. 2 of Forryan et al. (2013) [14].

It is necessary to choose the empirical constants in the above equations and in (6). There exists a broad literature discussing

these values for different laboratory and onsite conditions, but in the semi-empirical models they are only defined by scaling and can vary for different turbulent motions. We choose the outer turbulence scale based on the results of spectral approach Forryan et al. (2013); Galperin et al. (2007), in which the minimal wave number for the energy-carrying spectrum approximated by empirical functions is of the order of 2 cpm. Here we take $L = 0.58$ m. The range of the constant $C$ is also wide in the literature. Since we are mainly interested in the quasi-equilibrium regime when the shear source is balanced with dissipation

of turbulent energy, we use the data of Rodi (1980) to take $C = D = 0.09$. Anyway, we are only concerned about the order of obtained values; indeed, in the data of Forryan et al. (2013) considered below, the spread of data is about an order.

In what follows we solve the systems (2) and (3) using the Wolfram Mathematica 13 program and compare them with each other and the data of in situ measurements.

## 3   Application of the local model

As mentioned, here we apply the theory to the data of three cruises described in Forryan et al. (2013). First, we digitized red curves in Fig. 2 of that paper (as mentioned, there is a large dispersion of real data, but we naturally use mean profiles). Then, using the given profiles of $V_z$ and $N_2$, we calculated the Richardson number as shown in Fig. 1.

As seen from Fig. 1, Richardson number exceeds 1 within all range of the available data, and its maximum lies in the range of $10 - 23$.





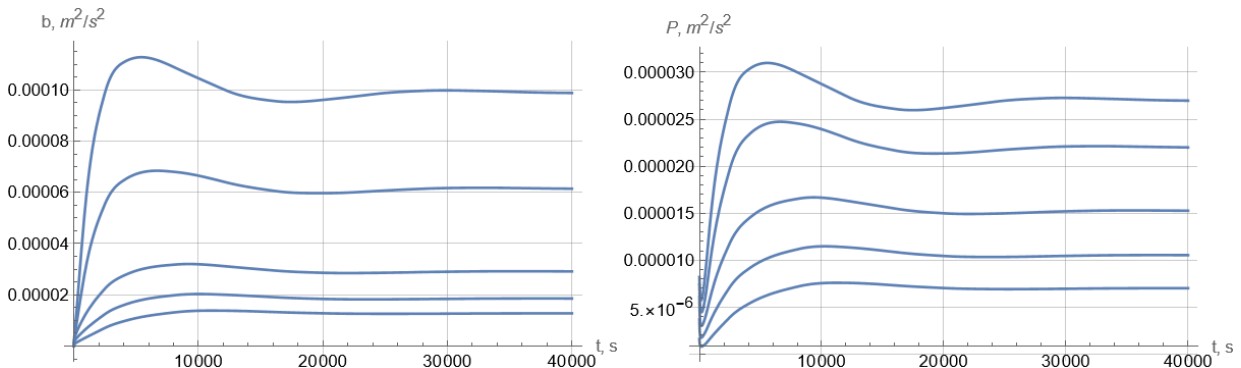

**Figure 2.** Temporal variation of kinetic (left panel) and potential (right panel) energy for the conditions of cruise JC29. From top to bottom: $z = 20, 30, 50, 100, 180$ m. Here $b_0 = P_0 = 10^{-6}$ m$^2$/s$^2$.

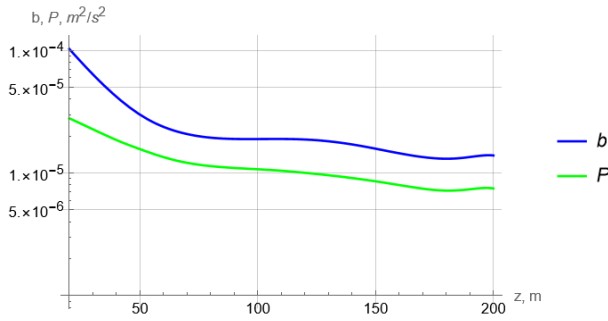

**Figure 3.** Profiles of kinetic (blue) and potential (green) energies at $t = 40000$ for JC209.

## 3.1 Cruise JC29

Now, using the interpolation of digitized data for $N^2$ and $V_z$ given in Fig. 2 of Forryan et al. (2013), we solved equations (3) with the initial conditions $b(0, z) = b_0 \exp(-0.1z)$ and $P(0, z) = P_0 \exp(-0.01z)$ (we remind that in the local model, $z$ is a given parameter). The values $b_0$ and $P_0$ varied from $10^{-6}$ to $10^{-5}$ m$^2$/s$^2$ with slight changes in transient processes but with the same asymptotic values of $b$ and $P$ at large times. Figure 2 shows the solutions for several depths covering the range shown in Forryan et al. (2013).

Here, the constant levels of energy are established in several hours. It is natural to use the asymptotic values for comparison with the measurement data. In the subsequent plots we use the log-lin presentation, following Forryan et al. (2013). Figure 3 shows the depth profile of the energies at large times. Note that, according to the second equation (4), the asymptotic values of $P$ and $b$ follow each other.

Using this solution, we calculate the turbulent dissipation rate (6) and compare it with the data of Forryan et al. (2013) after digitizing both and interpolating them by smooth functions. The result is shown in Fig. 4. Here the difference between

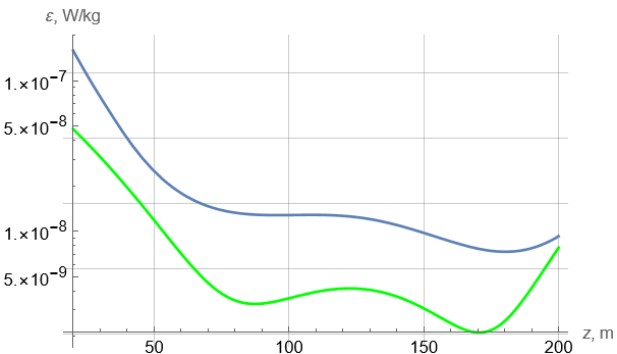

**Figure 4.** Profiles of turbulent kinetic energy dissipation rate for JC209. Green - interpolated data of Forryan et al. (2013) [14]. Blue - theory.

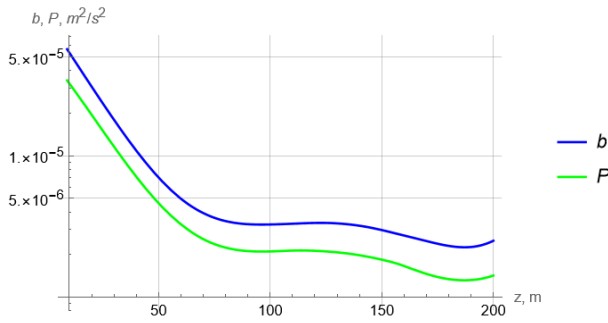

**Figure 5.** Profiles of kinetic (blue) and potential (green) at $t = 40000$ for D306.

theory and measurements is mainly within a half-order. Considering the large spread of experimental data, this is a rather good agreement.

## 3.2 Cruise D306

To save space, for another two cruises we show only the asymptotic profiles of the corresponding values at large times. For Cruise D306, the values of kinetic and potential energies are of the same order as for JC29 (Fig. 5).

Figure 6 shows the turbulence dissipation rate. Here again the difference between theory and data of Forryan et al. (2013) is within half-order.

## 3.3 Cruise D321

The corresponding dependencies for cruise D321 are given in Fig. 7 and 8. Here again, one can see a good agreement between the theory and the mean measured profile.

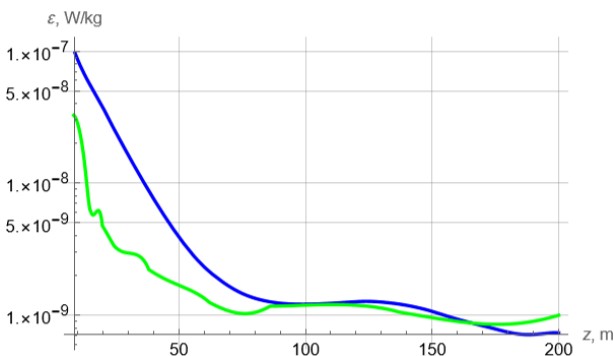

**Figure 6.** Profiles of turbulent kinetic energy dissipation rate for D306. Green - interpolated data of Forryan et al. (2013). Blue - theory.

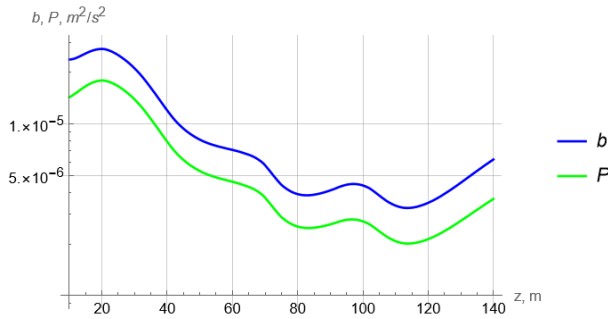

**Figure 7.** Profiles of kinetic (blue) and potential (green) at $t = 40000$ for D321.

## 4 Comparison with the full system

The above results were obtained in neglection of vertical turbulence diffusion. To verify this approximation, we solved the full system (2) with the same parameters and initial conditions, adding boundary conditions for fluxes of kinetic and potential energy:

$$Fb = \frac{\sqrt{b}\partial b}{\partial z}, \quad FP = \frac{\sqrt{b}\partial P}{\partial z} \tag{7}$$

in the form compatible with the initial conditions for initial points $z_0$ from which the plots start in Fig. 2 of Forryan et al. (2013), and zero values at the deepest points. Then the solutions for $b$ and $P$ were compared with those of the local system (3). Here we again show such a comparison for the asymptotic values. In what follows, the above local solutions are shown in blue, and the solutions of equations (2) with diffusion, in orange. In all three cases, the local and full models are practically identical. Evidently, this means the closeness of data for the dissipation rate that is a function of b. Hence, for the vertical scales of variation of average values, vertical diffusion can be neglected, and one can use the simplified local equations (3).


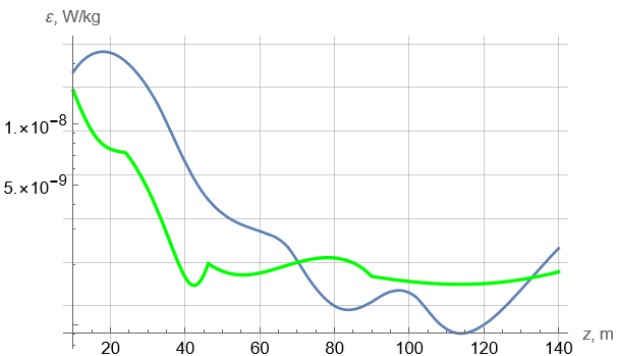

**Figure 8.** Profiles of turbulent kinetic energy dissipation rate for D321. Green - interpolated data of Forryan et al. (2013). Blue - theory.

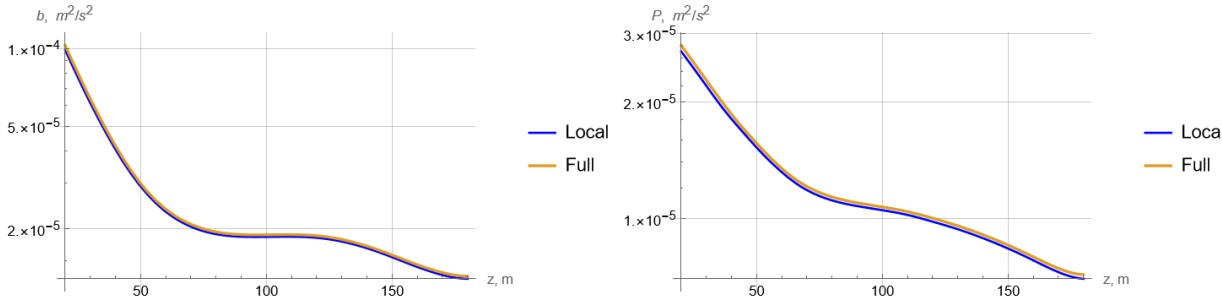

**Figure 9.** Cruise JC29: Comparison of profiles of kinetic (left panel) and potential (right panel) energies obtained from (3) (local) and (2) (full).

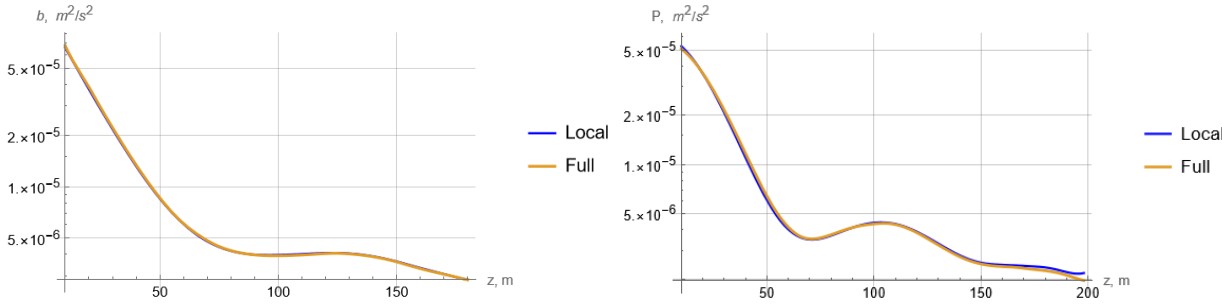

**Figure 10.** Cruise D306: Comparison of profiles of kinetic (left panel) and potential (right panel) energies obtained from (3) (local) and (2) (full).

## 5 Discussion and conclusions

In this paper we demonstrated that including the potential energy of turbulence (associated with density fluctuations in the
presence of stratification) in the semi-empirical, Reynolds-type equations of a turbulent flow allows to explain the existence and




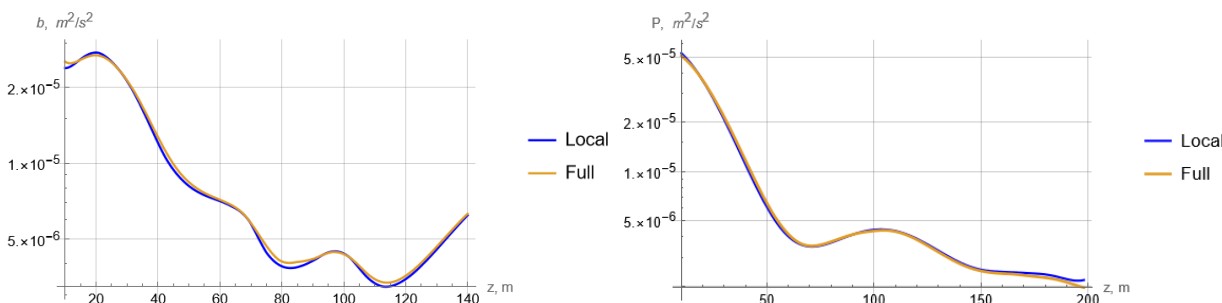

**Figure 11.** Cruise D321: Comparison of profiles of kinetic (left panel) and potential (right panel) energies obtained from (3) (local) and (2) (full).

evaluate the parameters of small-scale turbulence at large Richardson numbers. Application of these equations to the results of Forryan et al. (2013), where the measurements of profiles of buoyancy frequency, current shear, and dissipation rate of turbulent energy are shown together for three different areas of the ocean, provides not only qualitative but a reasonable quantitative agreement between the theory and experimental data. We have also shown that the contribution of turbulent diffusion to
the level of turbulent pulsations is insignificant. Thus, even in the conditions of strong stable stratification, the turbulence is maintained by the local shear of the mean flow velocity.

Here we limited ourselves by the semi-empirical approach with Kolmogorov scaling significantly modified by adding the equations for the potential energy of turbulence. Considering the dispersion of experimental data shown in Forryan et al. (2013) and a large variation of empirical parameters given in different sources (Smyth et al., 2013; Liu et al., 2017; You et al., 2003),
for now it seems more important to use the new, quantitative experimental data whenever available, rather than add more details to the relatively simple semi-empirical theory. In particular, we plan to extend the present approach to description of dynamic turbulence in the field of internal waves (e.g., Moum et al., 2022).

*Acknowledgements.* The work was supported by the RSF project No. 23-27-00002.





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

**Appendix A: Dynamical equations for a turbulent stratified flow**

Here we briefly outline the general system of equations for a turbulent stratified flow obtained in Ostrovsky and Troitskaya (1987) and developed in Soustova et al. (2020); Gladskikh et al. (2023). Without dwelling on the details which are described in these works, here we briefly outline the main points of the model. It begins by introducing the variable probability distribution function:

$$f(\boldsymbol{v}, \lambda, \boldsymbol{r}, t) = \langle \delta(\boldsymbol{u} - \boldsymbol{v}) \delta(\rho - \lambda) \rangle, \tag{A1}$$

where $\delta$ is Dirac delta-function, and the angular parentheses denote the ensemble averaging. Using this together with the Navier-Stokes equations for $\boldsymbol{u}$ and $\rho$, and supposing a Gaussian distribution function, the authors of Ostrovsky and Troitskaya (1987) obtained the expressions for the average fluxes of turbulent energy, momentum, and mass. They are the same as in the





common k-$\epsilon$ theory, except for the mass flux, having the form:

$$\langle \rho' u_i' \rangle = -LV\left( \frac{\partial \langle \rho \rangle}{\partial x_i} + g_i \frac{\langle \rho'^2 \rangle}{V^2 \rho_0} - \frac{\boldsymbol{g}\beta_i}{V^2 \rho_0} \right) \tag{A2}$$

where $V = \sqrt{\langle u'^2 \rangle}$ and $L$, as above, is the outer scale of turbulence, $g$ is the gravity acceleration, and $\beta_i$ are the components of the vector

$$\boldsymbol{\beta} = \frac{1}{4\pi} \int d\boldsymbol{r}_1 \frac{\partial}{\partial \boldsymbol{r}} \frac{1}{\left| \boldsymbol{r} - \boldsymbol{r}_1 \frac{\partial}{\partial z_1} \langle \rho'(\boldsymbol{r},t) \rho'(\boldsymbol{r}_1,t) \rangle \right|} \tag{A3}$$

which characterizes the effect of pressure fluctuations arising from random displacements of a particle in a stratified fluid.

The expression above for the mass flux includes the summand $g_i \frac{\langle \rho'^2 \rangle}{V^2 \rho_0} - \frac{\boldsymbol{g}\beta_i}{V^2 \rho_0}$, which, as shown below, leads to some significant differences from results obtained within the framework of known gradient models. The physical meaning of the additional terms in the expression mentioned above is related to the dependence of the force acting upon a random displacement of a liquid particle in a stratified medium on the shape of the liquid volume, namely, on the ratio of the characteristic scales $L_z$ and $L_r$. As shown in Ostrovsky and Troitskaya (1987) the components of the vector $\boldsymbol{\beta}$ have the form $\beta_x = \beta_y = 0$, $\beta_z =$

$\langle \rho'^2 \rangle R$ for a statistically homogeneous field of density fluctuations. Here $R$ is the anisotropy parameter:

$$R \simeq \begin{cases} 1, & L_z \ll L_r, \\ \simeq \left( \frac{L_r}{L_z} \right)^2 & L_z \gg L_r, \end{cases} \tag{A4}$$

where $L_z$ and $L_r$ are the vertical and horizontal scales of the density field correlation, respectively.

As a result, one obtains the equations for the mean values of velocity, density, turbulent kinetic energy $b = 3V^2/2$, and variance of density pulsations $\langle \rho'^2 \rangle$:

$$\frac{\partial \langle u_i \rangle}{\partial t} + \langle u_j \rangle \frac{\partial \langle u_i \rangle}{\partial x_j} + \frac{1}{\rho_0} \frac{\partial \langle p \rangle}{\partial x_i} + g_i \frac{\langle \rho \rangle - \rho_0}{\rho_0} = \frac{\partial}{\partial x_j} \left( L\sqrt{b} \left( \frac{\partial \langle u_i \rangle}{\partial x_j} + \frac{\partial \langle u_j \rangle}{\partial x_i} \right) \right), \tag{A5}$$

$$\frac{\partial \langle \rho \rangle}{\partial t} + \langle u_i \rangle \frac{\partial \langle \rho \rangle}{\partial x_i} = 2 \frac{\partial}{\partial x_i} L\sqrt{b} \left( \frac{\partial \langle \rho \rangle}{\partial x_i} + \frac{3}{2b\rho_0} \left( g_i \langle \rho'^2 \rangle + \boldsymbol{g}\beta_i \right) \right), \tag{A6}$$

$$\frac{\partial b}{\partial t} + \langle u_i \rangle \frac{\partial b}{\partial x_i} - L\sqrt{b} \left( \frac{\partial \langle u_i \rangle}{\partial x_j} + \frac{\partial \langle u_j \rangle}{\partial x_i} \right)^2 - \frac{\boldsymbol{g}}{\rho_0} L\sqrt{b} \left( \frac{\partial \langle \rho \rangle}{\partial z} + \frac{3\boldsymbol{g}}{2b\rho_0} \left( \langle \rho'^2 \rangle + \beta_z \right) \right) + \frac{Cb^{3/2}}{L} = \frac{5}{3} \frac{\partial}{\partial x_i} \left( L\sqrt{b} \frac{\partial b}{\partial x_i} \right), \tag{A7}$$

$$\frac{\partial \langle \rho'^2 \rangle}{\partial t} + \langle u_i \rangle \frac{\partial \langle \rho'^2 \rangle}{\partial x_i} - 2 \frac{\partial \langle \rho \rangle}{\partial x_i} L\sqrt{b} \left( \frac{\partial \langle \rho \rangle}{\partial x_i} + \left( g_i \langle \rho'^2 \rangle - \boldsymbol{g}\boldsymbol{\beta_i} \right) \frac{3}{2b\rho_0} \right) + \frac{Db^{1/3}}{L} \langle \rho'^2 \rangle = \frac{\partial}{\partial x_i} L\sqrt{b} \frac{\partial \langle \rho'^2 \rangle}{\partial x_i}. \tag{A8}$$

In an incompressible fluid considered here, $\nabla u = 0$. The potential energy of fluctuations is determined from the last equation

because of (A8). Equations (A6) are a particular case of this system for the given average current and density stratification. In general, such effects as internal wave damping by turbulence can be included in the solution as well. On the other hand, the turbulence "breakdown" phenomenon, in which, in certain phases of the wave, the velocity shear cannot maintain a nonzero level of turbulent energy obtained using the common semi-empirical equations (Ivanov et al., 1983), does not exist here. This is also confirmed by numerical calculations using parametrization obtained based on the model above, given in the work

(Gladskikh et al., 2023).