# Peer review of "Evolution of small-scale turbulence at large Richardson numbers"

_Nonlinear Processes in Geophysics, 2023_

## Referee Comment (RC2)

**Report on the manuscript npg-2023-22 entitled "Evolution of small-scale turbulence at large Richardson numbers" by Ostrovsky et al. submitted for publication to NPG**

The goal of this paper is the verification of the theory of the stably stratified turbulence for the oceanic flow using the data from the upper level oceanic turbulence. In particular, it was clearly demonstrated applying the data from the upper level oceanic turbulence that small-scale turbulence is maintained even at large gradient Richardson numbers. Main mechanism of this phenomenon is caused by conversion of turbulent kinetic energy into turbulent potential energy and self-control feedback of the increased fluctuations of density which decreases the vertical mass flux. Analogous phenomenon exists in the atmospheric turbulence where the self-existence of stably stratified turbulence is related to the conversion of turbulent kinetic energy into turbulent potential energy with increasing the vertical gradient of the mean potential temperature; and self-control feedback of the negative down-gradient turbulent heat transfer through efficient generation of the counteracting positive non-gradient heat transfer by turbulent potential energy.

This paper is very interesting and important. The topic of this paper is of a great interest to many readers of NPG. The presentation is clear and concise. The paper put the obtained results into context, with relevant references. The length of the paper is appropriate. The text is fluent and precise. The title and the abstract are pertinent and understandable to a wide audience. All figures are necessary and of appropriate quality. As a whole, the article contains new significant results and it reflects sufficiently high scientific standards to warrant its publication in NPG after minor corrections (see below):

1. In the left hand side of the second equation (2) for the turbulent potential energy $P$, the derivative $\partial P/\partial z$ needs to be replaced by $\partial P/\partial t$, where $t$ is the time and $z$ is the vertical coordinate.

2. Similarly, in the left hand side of the second equation (3) for the turbulent potential energy $P$, the derivative $\partial P/\partial z$ needs to be replaced by $\partial P/\partial t$.

3. Line 67: the bracket after $P$ in $Db^{1/2}P)/L$ needs to be removed.

4. In Eq. (4), $f(Ri$ needs to be replaced by $f(Ri)$.

---

## Author Response (AR1)

**Review 1**

The authors express their gratitude to the Reviewer for careful reading and high appreciation of the work. All typos have been corrected.

**Review 2**

The authors are grateful to the Reviewer for careful reading of the manuscript and valuable comments. The manuscript has been revised according to the comments. Please find below our responses to the main points of the review.

1. *The data analysis does not include any kind of uncertainty and statistical confidence estimates. The paper always presents just one curve for observations at each station. If we look into the paper by Forryan et al. (2013), which is referred in this study, we would see very scattered data there. The authors need to treat and account for this data scatter properly.*

Indeed, the data of Forryan et al., 2013 shows a significant, up to an order, data scatter from each cruise, obtained in different locations and on different days. Unfortunately, the authors of that paper did not specify confidence intervals of the data, and no correlation between different curves for shear and buoyancy frequency curves is known. All we could do is to evaluate the maximal possible scatter of the results. For that, we calculated maximal and minimal values of the Richardson number (by dividing the rightmost values of $N^2$ by the leftmost values of $S^2$ and vice versa) and found the corresponding extreme profiles for TKE dissipation rate. These maximal limits exceed the scattering for this value shown in Forryan, 2013 which implies that knowing the data for a specific location and time of the measurement, we could reasonably well predict the corresponding data scatter for turbulent kinetic energy dissipation rate. This plot for one cruise and the corresponding comments are added in the concluding section of the paper. The results for other two cruises are qualitatively similar.

2. *The text is written in understandable English, but many sentences are too long and cumbersome. E.g., the abstract consists of just 3 very long sentences, which are hard to understand. Moreover, the "Discussion and conclusions" section contains neither discussion nor conclusions. The authors do not discuss the results and did not state the conclusions, they just repeat what they did in the study.*

We made the abstract shorter. In the concluding section, after a summary of results, we added a discussion of data scatter (with the figure), as mentioned above. We also briefly specified the future problems. Hope we understood your comments correctly.

We also corrected minor flaws. In particular, the notation $b$ for TKE is replaced by $K$ (in alignment with $P$ for TPE).